# Facilitation of Motor Evoked Potentials in Response to a Modified 30 Hz Intermittent Theta-Burst Stimulation Protocol in Healthy Adults

**DOI:** 10.3390/brainsci11121640

**Published:** 2021-12-12

**Authors:** Katarina Hosel, François Tremblay

**Affiliations:** 1Bruyère Research Institute, Ottawa, ON K1N 5C8, Canada; khose083@uottawa.ca; 2School of Rehabilitation Sciences, Faculty of Health Sciences, University of Ottawa, Ottawa, ON K1H 8M5, Canada

**Keywords:** transcranial magnetic stimulation, motor evoked potentials, theta-burst stimulation, neuroplasticity

## Abstract

Theta-burst stimulation (TBS) is a form of repetitive transcranial magnetic stimulation (rTMS) developed to induce neuroplasticity. TBS usually consists of 50 Hz bursts at 5 Hz intervals. It can facilitate motor evoked potentials (MEPs) when applied intermittently, although this effect can vary between individuals. Here, we sought to determine whether a modified version of intermittent TBS (iTBS) consisting of 30 Hz bursts repeated at 6 Hz intervals would lead to lasting MEP facilitation. We also investigated whether recruitment of early and late indirect waves (I-waves) would predict individual responses to 30 Hz iTBS. Participants (*n* = 19) underwent single-pulse TMS to assess MEP amplitude at baseline and variations in MEP latency in response to anterior-posterior, posterior-anterior, and latero-medial stimulation. Then, 30 Hz iTBS was administered, and MEP amplitude was reassessed at 5-, 20- and 45-min. Post iTBS, most participants (13/19) exhibited MEP facilitation, with significant effects detected at 20- and 45-min. Contrary to previous evidence, recruitment of early I-waves predicted facilitation to 30 Hz iTBS. These observations suggest that 30 Hz/6 Hz iTBS is effective in inducing lasting facilitation in corticospinal excitability and may offer an alternative to the standard 50 Hz/5 Hz protocol.

## 1. Introduction

Theta Burst Stimulation (TBS) is a form of repetitive transcranial magnetic stimulation (rTMS) introduced in the mid-2000s by Huang et al. [1]. The original TBS protocol was based on animal studies showing that application of burst at a high rate (50–100 Hz) repeated at a low rate in the theta rhythm (4–7 Hz) induced long-term potentiation in the rodent’s motor cortex or hippocampus [2]. In their study, Huang et al. [1] demonstrated that a combination of 20 cycles of 50 Hz bursts repeated every 200 ms (i.e., 5 Hz) was effective in inducing lasting modulation in corticospinal excitability, as reflected in the amplitude of motor evoked potentials (MEPs). TBS tends to produce MEP suppression when delivered continuously for 40 s (600 pulses), whereas facilitation is observed when TBS is delivered intermittently (2 s ON, 8 s OFF) for 192 s. Following the original publication of Huang et al. [1], most subsequent TMS studies used the same combination of burst frequency (50 Hz) and inter-burst interval (5 Hz) to investigate TBS effects, this combination becoming some sort of ‘standard’ in the field [3,4].

While TBS protocols show promise as a therapeutic tool in neurological and psychiatric disorders [5], notably for symptomatic relief of major depression [6,7], their use in clinical settings remains limited by the considerable variability of responses both within and between individuals [8,9]. Among the many factors contributing to this variability, the use of non-optimal stimulation parameters (e.g., intensity, bursts, and inter-bursts frequency) has been pointed out as a contributing factor [3]. As stressed earlier, most investigators have relied on the 50 Hz/5 Hz standard to investigate TBS effects without considering whether such a pattern might be optimal. Only a minority of investigators have considered modifications to the ‘standard’ to determine whether altering TBS parameters could lead to more robust aftereffects. In this respect, Goldsworthy et al. [10], based on observations by Nyffeler et al. [11] regarding the effects of 30 Hz TBS on the oculomotor system, propose a modification to the original TBS protocol described by Huang et al. [1]. In their report, Goldsworthy et al. [10] showed that TBS delivered using a combination of 30 Hz bursts repeated at 6 Hz in the continuous mode evoked longer-lasting MEP suppression than 50 Hz/5 Hz protocol. Subsequent studies have provided further evidence regarding the effectiveness of the 30 Hz bursts in modulating corticospinal excitability [12,13,14]. However, much of this evidence has come from studies using the continuous mode, leaving the question of whether similar effects could be obtained with the intermittent mode. To our knowledge, only two studies reported on the effect of 30 Hz iTBS. Wu et al. [14] showed that 600 pulses of iTBS consisting of 30 Hz bursts repeated at 5 Hz intervals were effective in inducing MEP facilitation up to 10 min in healthy adults, while Pedapati et al. [15] made similar observations in children using the same iTBS parameters but for 300 pulses. Thus, while there is still limited data regarding the effects of 30 Hz bursts, the modified iTBS seems to be effective as the standard in modulating corticospinal excitability. In a recent systematic review of TBS effects, Chung et al. [3] concluded that, although there was evidence to suggest that 30 Hz TBS might produce more persistent and larger effects than 50 Hz TBS, more studies were required to validate its reliability.

In the present study, our goal was to seek further evidence for the effectiveness of 30 Hz TBS in inducing lasting modulation in corticospinal excitability. More specifically, we sought to determine whether the modified 30 Hz/6 Hz TBS protocol proposed by Goldsworthy et al. [10] would lead to lasting MEP facilitation when used in the intermittent mode. Our investigation also sought to determine whether individual differences in the recruitment of cortical interneurons in response to TMS would predict responses to 30 Hz iTBS, as reported by Hamada et al. [16]. To this end, we collected MEPs in response to anterior-posterior (AP), posterior-anterior (PA), and latero-medial (LM) stimulation to assess differences in MEP latency as an index of individual susceptibility to recruit early or late indirect waves (I-waves) in response to TMS.

## 2. Methods

### 2.1. Participants

Our initial recruitment targeted 30 participants based on a power analysis using the standardized mean difference of 0.71 for iTBS aftereffects reported by Chung et al. [3]. However, due to the COVID-19 pandemic restrictions, we could reach only 70% of our target. Thus, our sample consisted of 21 healthy adults (15 females; mean age, 25.3 ± 4.8 years; range, 19–40 years). All participants but three were right-handed, as determined with the Edinburg Hand Inventory. Before testing, participants were screened with a questionnaire to ensure they had no prior or current health conditions (e.g., multiple sclerosis, history of recent hand trauma or nerve injuries) that could interfere with our measures and for contraindications to TMS. The study procedures were approved by the institutional research ethics boards (Bruyère Protocol # M16-20-009; Ottawa Office of Research Ethics and Integrity, protocol# H-10-20-6523) and all participants provided written informed consent before participation. Because of the COVID-19 pandemic, participants were required to wear procedural masks during testing sessions to comply with mandatory safety procedures, while investigators were required to wear masks and visual shields.

### 2.2. Experimental Protocol

Figure 1 shows a schematic of the experimental protocol. Participants first underwent single-pulse TMS with the coil in the standard orientation (PA) to determine MEP amplitude at Time 0 (Baseline). Then, MEPs were elicited with the coil placed in the different orientations (i.e., AP, LM, PA) to assess differences in latency. Afterward, the 30 Hz iTBS protocol was administered. Within 5 min after iTBS, participants provided reports regarding tolerability and rated pain associated with the stimulation protocol with the visual analog scale (VAS). Then, MEPs were elicited at three specific time points post-iTBS (i.e., 5-, 20- and 45-min) to assess changes in corticospinal excitability:

### 2.3. Baseline Assessment of Corticospinal Excitability

Corticomotor excitability was assessed with participants seated in a recording chair with armrests. The right hand rested flat on a small wooden plate with two protruding rods to delimit the index and thumb finger position. Single-pulse TMS was delivered to the left hemisphere using a focal coil connected to a Magstim^®^ BiStim^2^ stimulator (Magstim Co., Whitland, UK). To assist in coil positioning, participants were fitted with a *Waveguard™* TMS compatible cap (ANT North America Inc., Philadelphia, PA, USA) and wore a U-shaped neck cushion to minimize head movements. The stimulation targeted the representation of the right first dorsal interosseus (FDI) muscle. The FDI hot spot was located by stimulating the approximate area on the left hemisphere at a relatively high intensity (e.g., 50% stimulator output) until MEPs could be evoked. The intensity was then decreased, and the area was further explored in 1-cm steps (anterior-posterior, mediolaterally) while stimulating to pinpoint the location. This site was then marked with a round 1-cm sticker to ensure consistent coil positioning. MEPs were recorded using Delsys surface sensors (DE-2.1, Boston, MA, USA). Amplification (gain = 1000) and filtering (bandwidth, 6–450 Hz) were performed via a Bagnoli™ 4 System (Delsys Inc., Boston, MA, USA). Electromyographic signals were digitized at a rate of 2 kHz via custom software on a PC equipped with an acquisition card (PCI-63203; National Instrument Corp., Austin, TX, USA). Each trial consisted of a 300 ms duration acquisition window with TMS pulses delivered at 150 ms. Resting and active motor thresholds (rMT, aMT) were determined using the Motor Threshold Assessment Tool software (MTAT 2.0; Clinical Researcher, Knoxville, TN, USA), which allows fast and reliable thresholds estimations with a minimal number of stimuli [17]. The MTAT software relies on the maximum likelihood strategy to estimate motor thresholds and involves a pre-set stimulation pattern with the assumption of response failure (MEP absent) for a subthreshold intensity and a success (MEP present) for a supra-threshold intensity. In this study, MEPs > 50 μV were used to determine response success for the rMT, whereas MEPs > 200 μV were used to determine response success for the aMT. At Time 0, baseline corticospinal excitability was assessed by applying single pulses with the BiStim^2^ at an intensity equivalent to 130% rMT with 5–10 s intervals between pulses.

### 2.4. Assessment of MEP Latency with Different Coil Orientations

Following the baseline assessment, single-pulse TMS was performed with the coil placed in three orientations to estimate the recruitment of direct and indirect waves (D-wave and I-waves) [16]. Before testing, the aMT was determined with the MTAT software while participants exerted a light static contraction (about 10% of their maximal) of the right FDI by pushing against the protruding rod with the index finger. The stimulator intensity was set at 110% aMT for MEPs elicited with the coil positioned in the standard PA orientation (i.e., the handle pointing 45° backward). For the AP orientation (handle pointing 45° forward) and LM orientations (handle pointing downward), the stimulation intensity was increased to 140% aMT to ensure recruitment of D-wave (LM stimulation) and late I-waves (AP stimulation). For AP and PA stimulations, 15 MEPs were recorded, whereas ten were recorded for LM stimulation. These numbers were deemed sufficient to provide a reliable estimate of the onset latency of MEPs [18]. The order of testing with the different coil orientations was counterbalanced across participants.

### 2.5. Modified 30 Hz/6 Hz iTBS Protocol

For iTBS, participants were moved to another chair to allow for the rTMS application. The 30 Hz iTBS was delivered using a Magstim® Rapid^2^ stimulator (Magstim Co., Whitland, UK) connected to a focal high-efficiency coil (D70^2^, Magstim Co.). Before application, the aMT was reassessed to account for the differences between stimulators and coils (i.e., BiStim^2^ monophasic pulses versus Rapid^2^ biphasic pulses). Once the aMT was determined, the stimulator intensity was set at 80% aMT in line with safety recommendations for TBS applications targeting the motor cortex [19]. The iTBS was delivered over the hand motor area and consisted of 10 trains of 30 Hz 3-pulse bursts applied at 6 Hz interval and repeated every 10 s (1.7 ON, 8.3 s OFF) for a total of 20 cycles (600 pulses over 192 s).

### 2.6. Post-iTBS Changes in Corticospinal Excitability, Safety and Tolerability

Following the iTBS protocol, participants were quickly returned to the recording chair for single-pulse TMS. During the time between the end of the iTBS session and the first post-iTBS time point, participants completed an rTMS adverse events questionnaire to assess safety and tolerability. Participants were asked to rate on a scale of 0 to 5 (none, minimal, mild, moderate, marked, severe) if they experienced any of the following symptoms after the intervention: headache, scalp pain, arm/hand pain, other pain, other sensations (e.g., tingling, burning), weakness, loss of dexterity, vision/hearing changes, ear ringing, nausea/vomiting, rash/skin changes, or others. The pain and discomfort associated with iTBS were also rated using the visual analog scale (VAS). At 5-, 20- and 45-min post iTBS, MEPs (*n* = 15) were elicited (130% rMT) to assess changes in corticospinal excitability.

### 2.7. Analysis of MEP Data

Analysis of MEP characteristics in terms of amplitude and latency was performed offline by the same investigator (KH) using custom software. MEPs were analyzed by first superimposing MEP traces recorded at each time point and testing condition. Then, mean peak-to-peak amplitude (mV) and latency (ms) were determined by visual inspection. Individual means for latency and amplitude were then reported in the database for further analysis. As mentioned earlier, individual susceptibility to recruit early and late I-waves in response to single-pulse TMS was assessed by computing the latency differences between MEPs recorded with AP stimulation and those recorded with LM or PA stimulation [16]. The latency difference was determined by subtracting the mean AP latency from the LM latency (i.e., AP-LM, but see below).

### 2.8. Analysis of Responses to iTBS

In line with previous studies [9,20], MEP amplitude was normalized to identify individuals who responded to the modified 30 Hz protocol. Specifically, responders and non-responders were operationally defined using a cut-off of ±10% from MEPs recorded at baseline (Time 0). MEP amplitude in mV recorded at each time point post-iTBS (i.e., Time 1, 2, and 3) was averaged to get a grand average. Then, MEP ratios were computed by expressing the grand average in percent relative to baseline (i.e., MEP_grand avg_/MEP_baseline_) x 100). Using the 10% cut-off, individuals showing facilitation (i.e., MEP ratio > 110%) were considered responders, while those showing either suppression (i.e., MEP ratio < 90%) or no modulation (i.e., 90% < MEP < 110%) were classified as non-responders.

### 2.9. Statistical Analysis

D’Agostino-Pearson’s test revealed that amplitude data at specific intervals post-iTBS were not normally distributed (Time 2, Time 3). As suggested by Nielsen [21], amplitude data were log-transformed to normalize the distributions. MEP log-amplitude data were then entered into a one-way repeated measure analysis of variance (ANOVA) with Time (0,1,2,3) as the repeated factor. Dunnett’s post-test was used for post hoc comparisons. The influence of biological sex was not considered in this analysis, for our sample of participants consisted mainly of females (13/19). Also, there is evidence that sex differences have little influence on neuromodulation induced by non-invasive brain stimulation protocols [22]. Latency data were normally distributed and did not need transformation. A one-way repeated measures ANOVA was performed on latency data to compare differences at the different coil orientations (AP, PA, and LM) using Tukey’s post-test for post hoc comparisons. Finally, a linear regression analysis was performed to determine whether latency differences predicted MEP modulation following iTBS. The level of significance was set at 0.05 for all tests. For ANOVA results, besides F and *p*-values, we also report partial eta squared (η^2^) as an index of the size of the intervention effect. All statistical tests and graphs were produced using GraphPad Prism version 9.0 for Windows™ (GraphPad Software, San Diego, CA, USA).

## 3. Results

### 3.1. Baseline Measures of Excitability and Latency Differences

Of 21 participants, 19 (13 females) completed the protocol without issues. Two female participants had to be excluded after experiencing minor adverse reactions (i.e., lightheaded, nauseous) to single-pulse TMS. At baseline, the average rMT was 44.1 ± 8.8%, and the mean MEP amplitude was 1.1 ± 0.8 mV. The average aMT, as determined with the BiStim^2^ stimulator, was 33.0 ± 5.7%. Figure 2a shows the distribution of latency values measured with the different coil orientations. As expected, participants exhibited shorter MEP latencies in response to LM stimulation when compared to either PA or AP stimulation (respective mean, 19.8, 20.8, 23.0 ms). The ANOVA confirmed that latencies differed significantly at the different coil orientations (F_2,36_= 22.3, *p* < 0.001, η^2^ = 0.55). Post-hoc comparisons indicated that latencies measured with LM and PA stimulation were significantly shorter than those measured with AP stimulation (Tukey’s post-test, *p* < 0.001). However, there was no difference between LM and PA stimulation (*p* = 0.19) (Figure 2a). The latter finding reflected the fact that some participants (*n* = 4) exhibited a shorter latency with PA than with LM stimulation. In those cases, the PA latency was used to compute the differences. The frequency distribution of latency differences (i.e., AP-LM/PA) computed across all participants is shown in Figure 2b. As evident in the figure, participants exhibited a relatively wide range of latency differences (1–7.5 ms) with a median difference at 3.5 ms.

### 3.2. Tolerability and MEP Modulation in Response to iTBS

Only mild adverse events were reported in association with the iTBS protocol. About three-quarters of the participants (14/19) experienced mild side-effects (ratings 1–3/5), mainly during the application in the form of scalp sensitivity (7/19), headache (6/19), and tingling or burning sensations (7/19). Most participants reported little to no pain (mean VAS score, 1.1 ± 1.5 cm), although one participant did report significant pain (VAS score, 6 cm). This elevated VAS score was likely related to the intensity used for iTBS in this participant who exhibited an unusually high aMT (67%).

Regarding MEP modulation, the distribution of individual MEP log-amplitude measured at each time point before and after iTBS is shown in Figure 3. It can be seen that MEPs tended to be enhanced post-iTBS with greater enhancement at 20 and 45 min. The ANOVA confirmed that Time (F_3,54_ = 4.3, *p* = 0.009, η^2^ = 0.19) had a significant effect on MEP amplitude with post-hoc comparisons pointing to significant differences from baseline (Time 0) at 20- and 45-min post (Dunnett’s *post*-test, *p* = 0.01 and *p* = 0.007, respectively).

### 3.3. Variability of Individual Responses

Although many participants exhibited the expected MEP facilitation post-iTBS, some variability was observed. This variability can be appreciated by inspecting Figure 4a, where individual changes in normalized MEP amplitude relative to baseline are shown across the different time points post iTBS. Of 19 participants, 68% (*n* = 13) were classified as responders (range, 112–388%), while the remaining 32% (*n* = 6) were classified as non-responders showing either suppression (*n* = 3, range, 65–73%) or no modulation (*n* = 3, range, 96–104%). Typical examples of MEP modulation in responders and non-responders following iTBS are shown in Figure 4b.

### 3.4. Latency Differences as Predictors of Responses to iTBS

Figure 5a shows the relationship between individual latency differences and corresponding normalized MEP amplitude in response to iTBS. This relationship was inverse, with large latency differences associated with no modulation or depression, while small ones were associated with facilitation. The linear regression analysis revealed that latency differences were significant predictors of responses to iTBS, accounting for 24% of the variance in MEP amplitude (r^2^ = 0.24, *p* = 0.03). To further examine the inverse nature of the association, participants were regrouped based on the median latency difference into an ‘early I-waves’ (*n* = 11, Difference < 3.5 ms) and a ‘late I-waves’ (*n* = 8, Difference > 3.5 ms) group [23,24]. As shown in Figure 4b, the early I-waves group tended to show larger MEP facilitation on average when compared to the late I-waves group. However, the difference was not significant when compared with the Mann-Whitney test (U = 32, *p* = 0.31), given the variability and the small number of observations in each group.

## 4. Discussion

In the present study, we sought further evidence regarding the effectiveness of a modified 30 Hz/6 Hz TBS protocol in the intermittent mode to induce lasting modulation facilitation of MEPs. Our results showed that the modified iTBS protocol effectively facilitated MEPs for up to 45 min post-stimulation. Further to this, our analysis of responders showed that these effects were relatively consistent, with more than two-thirds of the participants exhibiting the MEP facilitation. Our regression analysis also revealed that small latency differences were associated with facilitation, a finding contrasting with previous reports. In the following discussion, we will address the significance of these findings for the applications of iTBS protocols in experimental and clinical settings.

### 4.1. Corticospinal Excitability and Latency Differences at Baseline

At baseline, our group of participants exhibited the expected variations in rMT and MEP amplitude for adults in their age range (19–40 years). More specifically, both the average rMT (mean, 43%) and MEP amplitude (mean, 1.1 mV) were in line with previous reports on the reliability of measures of corticomotor excitability [18,23]. The range of latencies measured in our participants in response to stimulation at different coil orientations was comparable to that reported in previous studies [25,26]. The observation that some participants (4/19) exhibited a shorter latency with PA stimulation than with LM stimulation may have reflected individual differences at the anatomical or physiological level in the ability of TMS pulses to recruit D-wave or I_1_ wave [27]. At any rate, the observed range of latency differences (1–7.5 ms) corresponded with that reported by Hamada et al. [16].

### 4.2. Tolerability, MEP Modulation and Variability of Responses to 30 Hz iTBS

Regarding tolerability, the 30 Hz iTBS protocol was well tolerated by our group of participants, and, more importantly, no serious adverse events were reported. While two participants had to be excluded, these exclusions were related to vaso-vagal reactions after experiencing single pulse stimulation, which is uncommon but can happen in susceptible individuals [28]. We surmised that these reactions were partly attributable to the pandemic context and that participants had to wear masks during testing. Concerning the iTBS protocol, while many participants (74%) reported adverse events, these were generally mild and consisted of the expected side effects of rTMS applications (i.e., headache, scalp pain, and craniofacial discomfort). The overall level of pain perceived in association with the iTBS session was lower (mean, 1-cm) than that reported by Malm et al. [29] following 50 Hz/Hz iTBS in a group of clinically depressed patients (median VAS of 4 cm). However, in this study, iTBS targeted the prefrontal cortex for a total of 2400 pulses and at 90% rMT, which may have accounted for the higher pain ratings. In the present study, only one participant did report a high level of pain. As mentioned, this report was linked to high intensity of stimulation during iTBS, confirming that intensity is the main factor driving pain and discomfort during rTMS interventions.

Our analysis following the modified iTBS protocol showed that MEPs were facilitated for up to 45 min post-stimulation. The observation that significant facilitation was detected at 20 and 45-min post-iTBS and not at 5 min is consistent with a recent meta-analysis by Chung et al. [3]. In analyzing the results of 87 iTBS studies, these authors concluded that iTBS facilitatory effects on MEPs were more significant at mid-time points (20–30 min) than early time-points (<5 min) post-intervention. However, these authors also noticed that iTBS effects were more variable at later time points (i.e., >30 min post), which contrasts with the strong facilitation we detected at 45 min. On the other hand, another recent quantitative review by Wischnewski and Schutter [30] concluded that iTBS increases excitability for up to 60 min, consistent with our current observation. Regarding the magnitude of facilitation, on average, MEPs were enhanced by about 40% over baseline (mean 143%), an increase larger than that reported by Wischnewski and Schutter [30] in their quantitative review of iTBS effects. This observation reinforces our contention that the 30 Hz protocol elicited strong MEP facilitation. In agreement with this, Pedapati et al. [15] reported similar large effects (up to a 1.5-fold increase in MEP size) in children and adolescents in response to 300 pulses 30 Hz iTBS. Thus, in line with other recent reports on 30 Hz iTBS, our modified iTBS protocol seemed highly effective in eliciting lasting MEP facilitation with an overall increase in corticospinal excitability above the level reported in previous studies using the 50 Hz standard protocol.

Regarding variability, much like other iTBS reports, not every participant exhibited the expected facilitation following 30 Hz iTBS. As stated earlier, inter-individual variability has been a lingering issue in TBS studies for more than a decade now, with a growing number of studies reporting no change in cortical excitability or an “opposite” effect to what is expected [31]. To our knowledge, only one recent study has observed a similar rate of facilitatory responses (i.e., 68%; Guerra et al. [32] following standard iTBS. Most studies using the standard 50 Hz iTBS protocol have reported much lower response rates, including McCalley et al. [33], who recently reported only 33% of responders. It may be argued that high inter-individual response variability will persist regardless of the TBS protocol used in terms of bursting frequency and inter-burst intervals. For instance, protocols used to induce LTP and LTD in animal models are far more precise than rTMS protocols in the human scalp, which are more diffuse, leading to activation of large cortical networks comprised of a greater variety of cell types. Likewise, in vitro experiments on slices suggested a blurred line between LTP and LTD, as both responses can be induced using identical stimuli on different parts of the neuron or under different experimental conditions [34,35,36,37]. Thus, the variability of response to TBS and other rTMS protocols may reflect the natural properties of cortical networks and underlying physiological mechanisms [38,39,40,41]. A detailed understanding of these sources of variably could provide a basis for altered response to TBS in several neurological disorders. It will aid in designing more optimal interventions tailored to the individual.

### 4.3. Predictors of Responses to iTBS from Latency Differences

The present study found an inverse relationship between iTBS aftereffects in MEP modulation and latency differences. Participants with small latency differences tended to show MEP facilitation, while those with large differences tended to show suppression or no modulation. Such a relationship contrasts with the positive association reported by Hamada et al. [16], who found that the larger the latency difference and the greater likelihood of recruiting late I-waves, the greater the MEP facilitation in response to iTBS. Before interpreting this apparent contradiction, it is essential to emphasize that not all TBS studies have found the positive relationship reported by Hamada et al. [16]. For instance, Hinder et al. [9] found no association between large latency differences (i.e., >4 ms) and MEP facilitation following 50 Hz iTBS. In fact, in their report, 75% of the participants exhibiting MEP facilitation following iTBS exhibited small AP-LM latency differences (<4 ms), which is somewhat in line with the present observation linking MEP facilitation with small latency differences. More recently, Rocchi et al. [42], in exploring predictors of responses to cTBS, found no correlations between AP-LM latency difference and cTBS aftereffects. Thus, not all studies agree with the notion that preferential recruitment of late I-waves, as reflected in large AP-LM differences, are predictive of positive responses to iTBS. The inverse relationship we found between AP-LM/PA latency differences and MEP modulation suggests that preferential recruitment of early I-waves was likely an important factor in mediating the aftereffects of 30 Hz iTBS. Although speculative, it is conceivable that for the 30 Hz/6 Hz protocol, the recruitment and modulation of early I-waves might be more critical than for 50 Hz/5 Hz iTBS. In this respect, it is worth noting that the superiority of the 30 Hz over the 50 Hz TBS protocol was initially described for cTBS. Indeed, Goldsworthy et al. [10] showed that the 30 Hz cTBS protocol induced more significant and longer-lasting depression in MEPs. Given that the inhibitory effects of cTBS are thought to involve a reduction in the excitability of circuits generating early I-wave [43], it is tempting to suggest that 30 Hz/6 Hz combination might be more efficient in modulating early I-waves. Recruitment of early I-waves has also been implicated in other facilitation-inducing TMS paradigms. For instance, Di Lazzaro et al. [44] showed that modulation of I_1_ wave was critical in determining the magnitude of short-interval intracortical facilitation (SIFC), a form of facilitation observed when two TMS pulses at or above the threshold are delivered at interstimulus intervals of 1.5, 3 and 4.5 ms. Moreover, a recent study by Higashihara et al. [26] found that individuals exhibiting small AP-LM latency differences (<4 ms) also exhibited significantly higher SICF when compared to participants with large latency differences (>4 ms). These findings confirm that facilitatory effects are more likely to be expressed in individuals in whom recruitment of early I-waves is easily achieved via TMS. Interestingly, in the report of Hamada et al. [16], individuals who exhibited opposite responses to cTBS (i.e., MEP facilitation instead of depression) were also those that showed small AP-LM latency differences.

While recruitment of I-waves and individual susceptibility to TMS appears to be a significant factor in predicting TBS aftereffects, other factors might also be important. In fact, in our group of participants, differences in latency explained about 25% of the variance in MEP amplitude modulation, leaving a substantial proportion unexplained. Pharmacological studies suggest that the LTP-like aftereffects of iTBS [45] are linked with NMDA receptor-dependent glutamatergic transmission. One theory is that differences between individuals in baseline levels of glutamate and GABA, hence the balance between cortical excitation and inhibition, may contribute to varying responsiveness [46,47]. On this basis, the same NIBS paradigm, whether it be iTBS or other forms of rTMS, may result in variable responses, such that some individuals reach optimal levels of excitation while others show little to no effect. In addition, it has been suggested that the variable responses to TBS could be partly due to genetic factors [48]. Specifically, brain-derived neurotrophic factor *(BDNF)* polymorphism has been associated with measures of cortical plasticity [48,49,50,51,52,53,54], including both experience-driven and human cortical plasticity induced by iTBS [48,55]. Finally, other factors related to age differences, baseline excitability, and time of day have been identified as potential factors to predict TBS effects [8].

### 4.4. Study Limitations

This study presents certain limitations. Firstly, while our sample size was acceptable, a larger sample size would have been preferable, given the reported high variability of individual responses to TBS [8]. However, because of the COVID-restrictions, there were many barriers to recruiting research participants. Along the same line, the fact that our sample consisted mainly of female participants might have influenced our results since there is evidence that responses to rTMS interventions can vary across the menstrual cycle [22]. Our study protocol did not account for this possible confound for monitoring the menstrual cycle would have required hormonal testing, which was not easily available at the time of testing. Such monitoring is certainly a factor to consider for future studies. Second, our study protocol did not include a direct comparison with 50 Hz iTBS precluding any conclusion regarding the superiority of 30 Hz iTBS. While we acknowledge this limitation, one must consider again that this study was performed in the context of the worldwide pandemic, with restrictions on laboratory access and the amount of time research participants and experimenters were allowed to stay on-site. Also, there is already a large body of data regarding the effects of the standard 50 Hz TBS protocols on corticospinal excitability (see Chung et al. [3], for a review). The lack of a sham condition could be seen as another major limitation. However, our goal was not to test overall efficacy but rather to investigate the effectiveness of the modified 30 Hz/6 Hz iTBS protocol in inducing lasting MEP facilitation. Nevertheless, adding a sham condition could provide critical information regarding the influence of expectations and anticipation on individual responses to the modified iTBS protocol [33].

### 4.5. Conclusions

In conclusion, the present study investigated the effects of a modified 30 Hz iTBS protocol on corticospinal excitability. Our results showed that corticospinal excitability was increased for up to 45 min post-iTBS. Furthermore, these effects appeared less variable than those reported for the standard 50 Hz protocol, with more than two-thirds of the participants showing the expected MEP facilitation. Also, our regression analysis of latency differences as predictors of iTBS effects pointed to a different mode of action for the modified TBS protocol with modulation of circuits generating early, as opposed to late I-waves, as a preferential mechanism leading to MEP facilitation. Altogether, these results suggest that the modified 30 Hz/6 Hz iTBS might be a sound alternative to the standard protocol to induce lasting corticospinal facilitation. This finding may have implications for the applications of TBS interventions in clinical populations.

## Figures and Tables

**Figure 1 brainsci-11-01640-f001:**
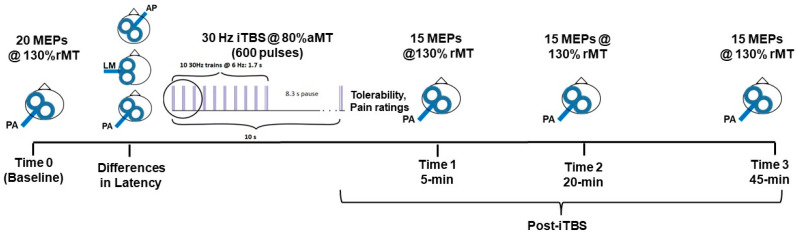
Schematic representation of the experimental protocol. Participants first underwent monophasic single-pulse TMS to assess corticospinal excitability at baseline. Then, single-pulse TMS was applied to assess latency differences for MEPS elicited with the coil placed in different orientations: Anterior-Posterior (AP), Posterior-Anterior (PA), and Latero-Medial (LM). Afterward, participants received the modified 30 Hz/6 Hz intermittent theta-burst protocol (iTBS, 600 pulses, intensity 80% of the active motor threshold (aMT). Changes in corticospinal excitability after iTBS were measured at specific post-intervention times: 5-, 20- and 45-min. The intensity used to test corticospinal excitability at baseline and post-iTBS was set at 130% of the resting motor threshold (rMT).

**Figure 2 brainsci-11-01640-f002:**
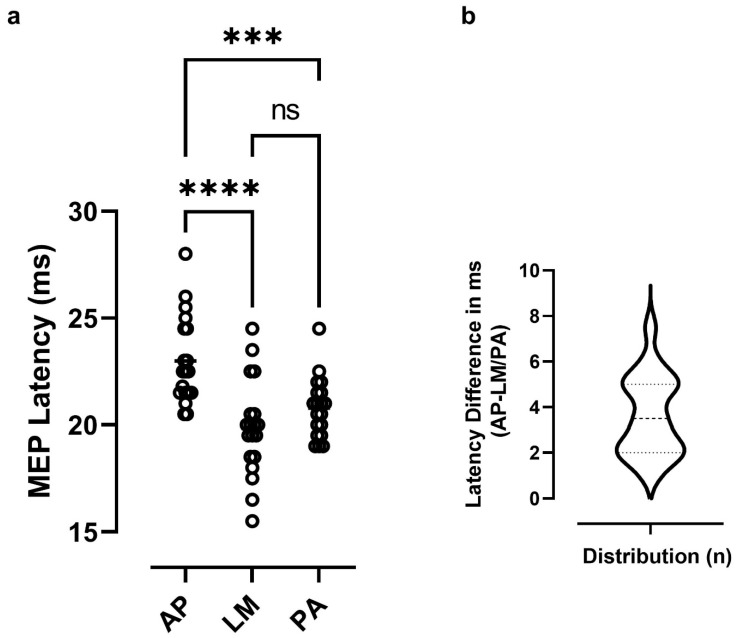
(**a**) Distribution of MEP latencies observed at the different coil orientations. Note that latencies measured using the AP orientation were significantly longer than those measured with either the LM or PA orientation (*** *p* < 0.001, **** *p* < 0.0001). (**b**) Violin plot illustrating the frequency distribution of latency differences (AP-LM\PA) computed in all participants (**b**). The dashed line in the plot corresponds to the median (3.5 ms), while the upper and lower dotted lines correspond to the quartile.

**Figure 3 brainsci-11-01640-f003:**
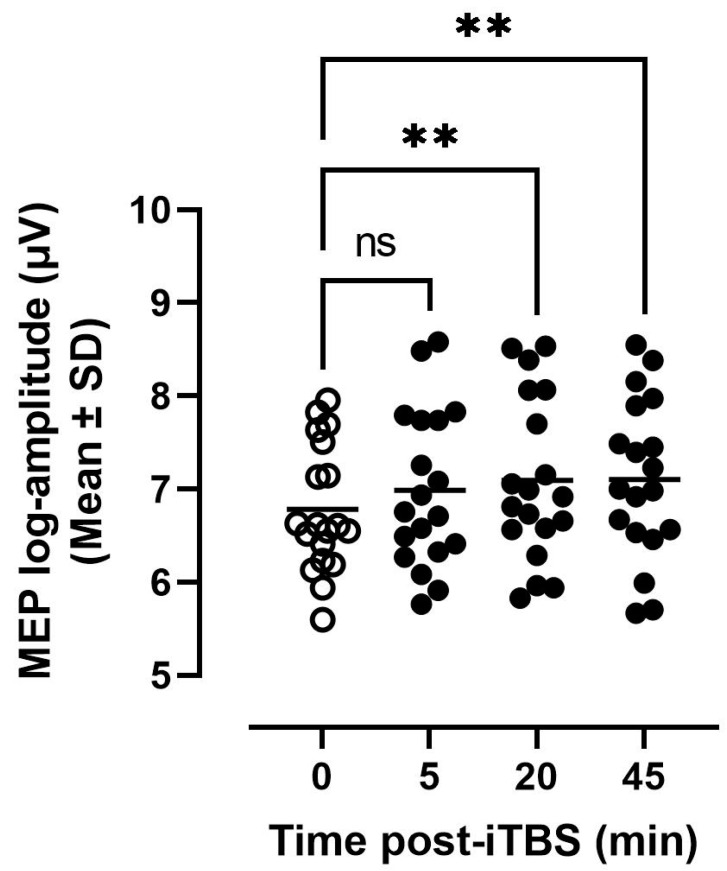
Distribution of individual MEP log-amplitude measured across the different time points relative to iTBS application. MEP-log amplitude measured at 20- and 45-min post-iTBS were significantly different (** *p* < 0.01) from those measured at baseline (i.e., Time 0).

**Figure 4 brainsci-11-01640-f004:**
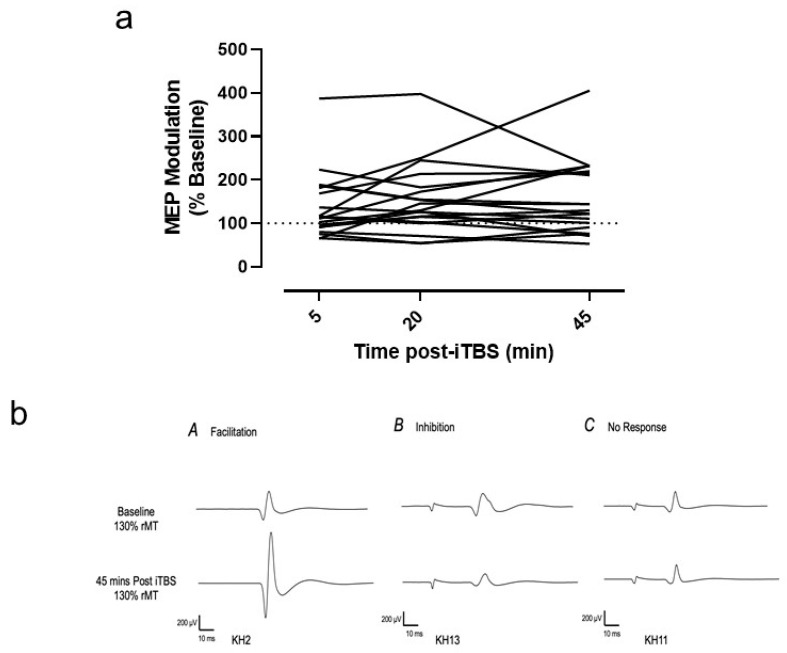
(**a**) Individual changes in MEP amplitude, when normalized relative to baseline, across the different time points post-iTBS. (**b**) Examples of MEP modulation recorded in response to 30 Hz iTBS. Facilitation (MEP > 110%) was observed in most (13/19) participants, while a minority exhibited either suppression (MEP < 90%, *n* = 3) or no modulation (90 < MEP < 110, *n* = 3).

**Figure 5 brainsci-11-01640-f005:**
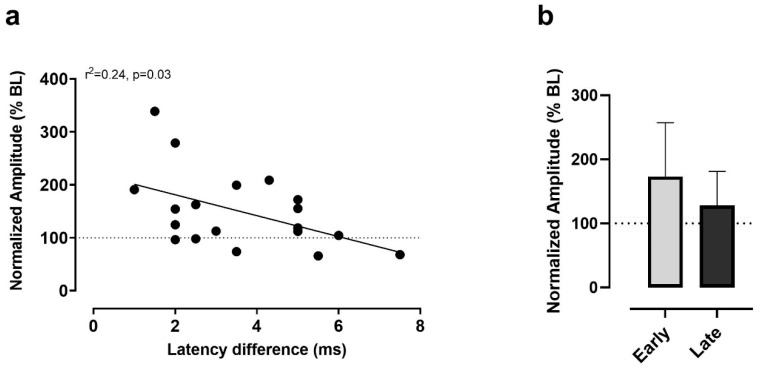
(**a**) Relationship between latency differences measured in participants and corresponding normalized MEP amplitude in response to iTBS. (**b**) Comparison of MEP amplitude modulation post-iTBS in participants exhibiting ‘Early’ (*n* = 9) versus ‘Late’ I-waves recruitment. The two groups were split based on the median latency difference (i.e., Early group, differences < 3.5 ms; Late group, differences > 3.5 ms).

## Data Availability

The data presented in this study are available upon reasonable request from the corresponding author.

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
