# Peer review of "Facilitation of Motor Evoked Potentials in Response to a Modified 30 Hz Intermittent Theta-Burst Stimulation Protocol in Healthy Adults"

_brainsci, 2021, doi:10.3390/brainsci11121640_

Round 1
Reviewer 1 Report
The present study entitled “Facilitation of motor evoked potentials in response to a modified 30 Hz intermittent theta-burst stimulation protocol in healthy adults” investigated the effectiveness of intermittent 30 Hz/6 Hz TBS in inducing lasting facilitation of MEPs. The authors reported that the modified iTBS protocol facilitated MEPs for up to 45 min post-stimulation and that small latency differences were predictive of the facilitatory effect on corticospinal excitability. The study is interesting and well written. In my opinion, the major potential issues were already discussed in the limitation sections. I only have minor comments and questions that the author should answer to improve the manuscript.
Introduction
Line 31 – "(4-10 Hz) induced…" this is not theta range; please correct.
Line 32 – add the reference.
Methods
Participants
Did the authors take note of the menstrual cycle of the female participants?
Although the differences in handedness would not profoundly affect the overall results, can the author check if there were significant differences in stimulation intensity (AMT and RMT) in right and left-handed participants? Left-handed individuals often have a high threshold on the left motor cortex.
Line 89-90: edit this part.
Line 145 - Edit the subheading
Results
Are there differences in stimulation intensities taken from the two devices (Magstim Rapid and BiStim2)?
Can you report the F value and effect size? Would it be interesting if the present data can give the same effect size (0.71) as those reported by Chung et al. (2016)?
In line 209, the authors reported that “Of 21 participants, 19 (13 females) completed the protocol without any issues. Two female participants had to be excluded….”. However, looking at Figure 3, it is obvious that plotted data were from 20 participants. Can the author correct this discrepancy?
Is the cut-off plus or minus 10 standard for the responder and non-responders? Looking again at Figure 3, we can observe very high variability. In my opinion, plus or minus 10% is arbitrary.
Although we would assume that the figure on the left is always the letter “A” and the one on the right is “B”, for the sake of clarification, kindly add a letter legend on the figures.
Discussion
Line 402: in SIFC, one pulse is suprathreshold, and the other is subthreshold.
Author Response
Reply to Reviewer#1
Introduction
Line 31 – "(4-10 Hz) induced…" this is not theta range; please correct.
Thanks for pointing out the error in the range. Correction made in the text.
Line 32 – add the reference.
Reference added.
Methods
Participants
Did the authors take note of the menstrual cycle of the female participants?
No, we did not. We have added a statement in the study limitations to address the issue.
Although the differences in handedness would not profoundly affect the overall results, can the author check if there were significant differences in stimulation intensity (AMT and RMT) in right and left-handed participants? Left-handed individuals often have a high threshold on the left motor cortex.
The three left-handed participants exhibited rMT and aMt typical of young adults of their age (Right, n=16, mean rMT 42%, aMT, 32%, see below for Left). We did not perform bilateral testing, so we cannot comment on hemispheric differences in excitability. However, our own investigation on the matter showed that Left-handers had higher rMT on the right hemisphere and, in fact, those individuals exhibited comparable rMT on the left hemisphere to that of mixed and right-handed individuals ( See Figure 3 in Davidson & Tremblay 2013)
Subject Left-handed rMT aMT
KY13 56% 46%
KY8 36% 34%
KY19 54% 38%
- Davidson, T., and F. Tremblay. "Hemispheric Differences in Corticospinal Excitability and in Transcallosal Inhibition in Relation to Degree of Handedness." PloS One 8, no. 7 (2013): e70286.
Line 89-90: edit this part.
The statement has been revised. We think it is important for readers to know that participants wore masks during testing because of the pandemic.
Line 145 - Edit the subheading
Correction made.
Results
Are there differences in stimulation intensities taken from the two devices (Magstim Rapid and BiStim2)?
Yes, there are some differences given the difference in pulse configuration and coil output, However, rMT and aMT with the two stimulators were highly correlated at the individual level (r2=0.55).
Can you report the F value and effect size?
We have revised the statistical analysis given the Reviewer’s comment below. Thanks to the Reviewer, we found out that there was a duplicate value in the dataset for MEP amplitude (hence the issue with 20 subjects in Figure 3). The error has been corrected and the analysis redone. Note that the correction did not change the results or the conclusions. We now report for the ANOVA’s the F value and the partial eta squared to indicate the size of the intervention.
Would it be interesting if the present data can give the same effect size (0.71) as those reported by Chung et al. (2016)?
Indeed, it would be interesting. The size of the intervention for variations in MEP amplitude, as reflected in eta squared, is 0.19, which appears smaller than that reported by Chung et al. However, such an effect size could hardly be compared with the “standardized mean difference’ computed by Chung et al. The SMD is based on the assumption that differences in standard deviations among studies reflect differences in measurement scales and not real differences in variability among study participants (which is the case here)..
In line 209, the authors reported that “Of 21 participants, 19 (13 females) completed the protocol without any issues. Two female participants had to be excluded….”. However, looking at Figure 3, it is obvious that plotted data were from 20 participants. Can the author correct this discrepancy?
Please see above our reply regarding F values.
Is the cut-off plus or minus 10 standard for the responder and non-responders? Looking again at Figure 3, we can observe very high variability. In my opinion, plus or minus 10% is arbitrary.
We agree it does seem arbitrary but this approach has been used by many other investigators (see Hinder et al or Perellon et al) to assess neuroplastic responses to rTMS. In this respect, the ±10% cutoff provides a reasonable criterion to classify responders given the variability of responses to rTMS interventions.
- Hinder, M. R., E. L. Goss, H. Fujiyama, A. J. Canty, M. I. Garry, J. Rodger, and J. J. Summers. "Inter- and Intra-Individual Variability Following Intermittent Theta Burst Stimulation: Implications for Rehabilitation and Recovery." Brain Stimulation 7, no. 3 (2014): 365-71.
- Perellon-Alfonso, R., M. Kralik, I. Pileckyte, M. Princic, J. Bon, C. Matzhold, et al. "Similar Effect of Intermittent Theta Burst and Sham Stimulation on Corticospinal Excitability: A 5-Day Repeated Sessions Study." European Journal of Neuroscience 48, no. 4 (2018): 1990-2000.
Although we would assume that the figure on the left is always the letter “A” and the one on the right is “B”, for the sake of clarification, kindly add a letter legend on the figures.
Done
Discussion
Line 402: in SIFC, one pulse is suprathreshold, and the other is subthreshold.
We have corrected the sentence to state that SICF is observed when two stimuli are given at or above threshold.
Reviewer 2 Report
This study investigated whether the modified 30 Hz/6 Hz TBS protocol (intermittent mode) proposed by Goldsworthy et al. (2012) would lead to lasting MEP facilitation. Some questions please need authors to answer.
- Why did not compare to the sham stimulation group and 50 Hz/ 5 Hz standard iTBS protocol? I think COVID-19 is not reason. Do you think 30 Hz/6 Hz TBS protocol (intermittent mode) increased MEP amplitude as compared to 50 Hz/ 5 Hz standard iTBS? What are the benefits of 30 Hz/6 Hz TBS protocol (intermittent mode) in comparison with 50 Hz/ 5 Hz standard iTBS?
- Does the participants have a neurological disease or other diseases? Please write in the manuscript.
- The single-pulse TMS elicited 20 MEPs at baseline but at three specific time points after post-iTBS (i.e., 5-, 20- and 45-min) just elicited 15 MEPs respectively. Why did not also test 20, 25 or 30 MEPs?
- In figure 1, the rMT needs to be defined.
- In line 125, MT needs to be defined.
- Please do not define aMT the second time in line 134.
- In line 145, full stop marked an error.
- In line 28, please define rTMS.
- How do you test the aMT and rMT? Please give the detailed process.
- How do you get the hotspot? Please give the detailed process.
- In lines 459 and 461, pay attention to the space between words.
Author Response
Reviewer #2
1.Why did not compare to the sham stimulation group and 50 Hz/ 5 Hz standard iTBS protocol? I think COVID-19 is not reason. Do you think 30 Hz/6 Hz TBS protocol (intermittent mode) increased MEP amplitude as compared to 50 Hz/ 5 Hz standard iTBS? What are the benefits of 30 Hz/6 Hz TBS protocol (intermittent mode) in comparison with 50 Hz/ 5 Hz standard iTBS?
We agree with the Reviewer that having a direct comparison with the 50 Hz protocol would have been ‘ideal’ but we have clearly stated the reasons as to why we did not perform such a comparison. First, our primary goal was to describe the effects of a modified 30 Hz iTBS protocol and not to test the relative efficacy of 30 vs 50 Hz bursts. Besides, as we have stated, there is already ample evidence of the effects of 50 Hz/5 Hz in the literature. Second, as we have explained in the limitations, our study was performed in the context of a worldwide pandemic. In this respect, we disagree with the reviewer. The pandemic has and continues to have a major impact on our research activities. Our lab is located in a geriatric hospital setting. For this reason, we had to follow very strict rules regarding who was coming on the premises and how long they would stay. We were allowed to recruit research participants only if they came for a short period and stayed in one room. A comparison between 30 and 50 Hz protocols would have meant having participants come twice (on different days), exposing them each time to risks for COVID. As for the benefits of the 30 Hz protocol, those are clearly stated in the discussion regarding the number of responders and the size of the facilitation.
- Does the participants have a neurological disease or other diseases? Please write in the manuscript.
We have added a sentence to clarify that participants were screened for health conditions.
3 The single-pulse TMS elicited 20 MEPs at baseline but at three specific time points after post-iTBS (i.e., 5-, 20- and 45-min) just elicited 15 MEPs respectively. Why did not also test 20, 25 or 30 MEPs?
We collected more MEPS at baseline to account that variations in excitability are often larger when participants received their first series of TMS pulses (even if they have received some stimuli before collecting data to experience the sensation). Thus, 20 MEPs were deemed adequate to get a reliable estimate of resting corticomotor excitability. Along with the same reasoning, we collected 15 MEPs post iTBS since participants were now more familiar with the sensation elicited by TMS pulse and saved time. As we have explained, we had constraints imposed on us by the local REB to limit the time participants were on-site. The fact that we use 130% rMT for the test intensity also reduced inter-trial variability to provide reliable estimates of corticospinal excitability.
- In figure 1, the rMT needs to be defined.
Done
- In line 125, MT needs to be defined.
We have revised the sentence so that abbreviations are defined.
- Please do not define aMT the second time in line 134.
Correction done.
- In line 145, full stop marked an error.
Correction made.
- In line 28, please define rTMS.
Done
- How do you test the aMT and rMT? Please give the detailed process.
We have provided more explanations in the text as to how we used the MTAS software to estimate MT.
- How do you get the hotspot? Please give the detailed process.
We have added more details as to how the hotspot was determined?
- In lines 459 and 461, pay attention to the space between words.
Corrections were made to check for extra spaces.
Round 2
Reviewer 2 Report
- In line 119, the capital "a" changes lower case.
- In line 226, notice the format of the comma.
- In line 253, where is the F value?
- In line 282, the result showed r2 = 0.24, p = 0.03 but it is a difference in Fig. 5.
- Although you state you defined the rMT in Fig. 1 you did not.
Author Response
- In line 119, the capital "a" changes lower case.
- Correction done
- In line 226, notice the format of the comma.
- Correction done
- In line 253, where is the F value?
- Correction done. The F value has been added.
- In line 282, the result showed r2 = 0.24, p = 0.03 but it is a difference in Fig. 5.
- The coefficient of determination is for the regression analysis and indicates that Lat differences were significant predictors of MEP facilitation. However, as we explained in the text, when participants were sorted into two groups (early and late I-waves), the difference between the two was not significant according to the Mann-Whitney U test.
- Although you state you defined the rMT in Fig. 1 you did not.
- Actually, we did, but it seems that the new figure legend with the correction was not properly inserted when the system reformated the document.